# COVID-19 pandemic impact on dentists in Latin America's epicenter: São-Paulo, Brazil

Tatiane Fernandes Novaes[1], Maisa Camillo Jordão[1], Carlos Felipe Bonacina[1], André Oswaldo Veronezi[1], Carlos Ariel Rodrigues de Araujo[1], Isabel Cristina Olegário[2], Daniele Boina de Oliveira[3], Veranika Ushakova[4], Alexander Birbrair[4], Danielle da Costa Palacio[3,5], Debora Heller[1,3,6]*

1 Postgraduate Program in Dentistry, Cruzeiro do Sul University, São Paulo, Brazil, 2 Dublin Dental University Hospital, Trinity College Dublin, Dublin, Ireland, 3 Hospital Israelita Albert Einstein, São Paulo, Brazil, 4 Department of Pathology, Federal University of Minas Gerais, Belo Horizonte, Minas Gerais, Brazil, 5 School of Dentistry, Departamento de Odontologia Social, Universidade Estadual de Campinas, Piracicaba, Brazil, 6 Department of Periodontics, University of Texas Health Science Center at San Antonio, San Antonio, Texas, United States of America

* debora.heller@cruzeirodosul.edu.br

**Data Availability Statement:** All relevant data are within the manuscript and its Supporting Information files.

## Abstract

The state of São Paulo, Brazil, where more than 94.000 dentists are currently registered, has become the epicenter of COVID-19 in Latin America. The aim of this cross-sectional study was to evaluate the impact of COVID-19 pandemic on dentists in this state. A semi-structured questionnaire was sent via e-mail to 93.280 dentists with active registration in the Dental Council of São Paulo (CROSP). The impact of COVID-19 pandemic was assessed through questions related to demographic, socioeconomic, dental practice characteristics and personal protective equipment (PPE) use. Ordinal logistic regression analysis was performed to investigate the association between all the variables (p<0.05). Over 8 days, 2113 responses were received. Only 26.52% of the sample reported a low-income reduction (from 0–10%), while the majority of dentists reported a more negative financial impact, 35.6% with a reduction of more than 50% of their monthly income. Dentists who worked in the private sector and at the capital had a greater financial impact when compared to those of the public sector and countryside of the state (p<0.05). Furthermore, about 83% reported not having received any specific training to control the transmission of coronavirus in the health area. This study provides evidence of the negative impact of the COVID-19 pandemic on the routine of dentists in the state of São Paulo, Brazil. Hopefully, this study will help dental and other health care professionals to better understand the consequences of disease in dental settings and strengthen preparedness throughout the dental health care system.

## 1. Introduction

In early December 2019, an outbreak of pneumonia with an unknown cause emerged in Wuhan, Hubei (China). The clinical symptoms were similar to a viral pneumonia, including fever, dizziness, and cough [1]. After sequencing analysis of samples obtained from the respiratory tract, the pathogen was identified as a new coronavirus (coronavirus 2, SARS-CoV-2,

**Funding:** This research was funded by Instituto Serrapilheira, grant number Serra-1708-316 15285 (AB)-https://serrapilheira.org/en/; Pró-reitoria de Pesquisa/Universidade Federal de Minas Gerais, grant number 05/2016 (AB)-https://www.ufmg.br/prpq/; FAPESP, grant number 2013/07467-1 and 2017/17943-6 (AB)-https://fapesp.br/. The funders had no role in study design, data collection and analysis, decision to publish, or preparation of the manuscript.

**Competing interests:** The authors have declared that no competing interests exist.

COVID-19), causing a severe acute respiratory syndrome due to phylogenetic similarity with SARS-CoV [2]. COVID-19 infected cases have since grown exponentially, and as a result, the World Health Organization (WHO) declared a public health emergency of international interest [3]. There are more than 68 million cases confirmed worldwide to date, with more than 1,5 million deaths [4]. SARS-CoV-2 was first reported in Brazil on February 25th, 2020, and due to its exponential spread, the country has become the epicenter for COVID-19 in Latin America, with more than 6,6 million cases and more than 177 thousand deaths [4].

Considering that the virus is transmitted predominantly through aerosol and salivary droplets [5] and that dental interventions are performed close to the oropharyngeal region, dentists are exposed to a high risk of infection [6]. In fact, due to this risk of widespread transmission of SARS-CoV-2 in dental practice, a suspension of elective care was imposed by national and international dental health entities [7] and new management protocols for dental practitioners have been released [8]. As a consequence, dentists worldwide reported seeing fewer patients, and an increase in the expenses on personal protective equipment (PPE). The higher demand for PPE has caused supplies shortages, including N95 face masks, that have become mandatory when dealing with aerosol procedures [9, 10], contributing to the vulnerability of all dental care professionals.

The state of São Paulo, Brazil is considered the financial heart of Brazil and has become the epicenter of COVID-19 in Latin America [11]. Currently, there are more than 94.000 dentists registered in the local Dental Council [12]. Thus, our aim was to evaluate the impact of COVID-19 pandemic on dentists in the state of São Paulo, Brazil. The null hypothesis of the study is that the COVID-19 pandemic would not impact the routine of dentists in the state of São Paulo, Brazil. The study's alternative hypothesis is that the COVID-19 pandemic would impact the routine of dentists in the state of São Paulo, Brazil.

## 2. Methods

### 2.1. Ethical aspects

This work was conducted according to the Helsinki declaration. The project was previously approved by the ethics committee of the Universidade Cruzeiro do Sul under CAAE number: 31720720.9.0000.8084. In addition, all methods were performed in accordance with the relevant guidelines and regulations and the original datasets generated and/or analyzed during the current study are available from the corresponding author on reasonable request. The consent form was made available prior to the questionnaire, in writing and consent was given through an alternative in the online questionnaire. Only dentists who read the informed consent form and agreed to participate, were included in the study.

### 2.2. Study design and data collection instrument

The STROBE checklist was [13] followed to show transparency in the analysis and reporting of this observational study (S1 Table). In addition, SURGE guideline (The SUrvey Reporting GuidelinE) was used for reporting the methods and results of the present survey [14].

A cross-sectional study was designed and a semi-structured questionnaire was sent to all dentists with active registration in the Regional Dental Council of São Paulo (CROSP), Brazil, who have email addresses cataloged in their database. In addition, an Instagram disclosure and WhatsApp messages were also sent in order to enhance dentist's participation and encourage them to check their e-mail inbox.

In order to certify a good reliability and validity of the questionnaire and its dimensions, we conducted a pre-test in a sample of 10 dentists. The pre-testers were asked to answer the questionnaire, and give their feedback on its clarity and record the time spent to complete all

questions. The authors discussed all questions until reaching a consensus. Pre-testers were not eligible to participate in this study, since they were dentists from public and private sectors from others states in Brazil.

### 2.3. Data collection instrument

The questionnaire was hosted online (Google Forms) during 8 days, from June 25, 2020 until July 02, 2020, and consisted of 26 questions, divided into 3 main groups: Dimension 1 –Socio-demographic characteristics of the interviewed population (8 questions), including its inclusion or not in SARS-Cov-2 risk groups; Dimension 2—Education characteristics of the interviewed population, including information/courses regarding safety protocols for clinical practice (8 questions); Dimension 3 –Clinical/Work characteristics and economical variables (10 questions) The original questionnaire and a translation of the questionnaire are presented in S2 and S3 Tables.

### 2.4. Statistical analysis

Data were analyzed using Stata/SE 16.11 statistical software (StataCorp LP, College Station, Texas, USA). Descriptive analysis was used to describe general, socioeconomic and dental practice characteristics of the interviewed population (n/%). An alluvial diagram was built using RAWGraphs software to compare the distribution between workplace (public/private/both) with financial impact. Unadjusted and adjusted ordinal logistic regression analysis were performed to investigate the association between the financial impact (income reduction: 0–10%; 10–50%; >50% reduction) and independent variables. Each answer to the questionnaire was considered an independent variable and all of them were considered in the adjusted model. Forwards stepwise ordinal logistic regression was performed using a p-value <0.20 as the criteria for inclusion in the adjusted model, and only variables with a p-value <0.05 were kept in the final model. Estimates of odds ratio are shown with 95% confidence intervals.

### 3. Results

From total of 93.280 dentists invited to participate by e-mail, 2,348 accessed the Google forms link provided and from those, 2.113 accepted to participate after reading the informed consent form. About 10% (n = 235) opted to not participate in the present study.

Dentists from the countryside (43%) and from the capital (57%) answered the questionnaire. The majority of the respondents were female (n = 1,565; 74%) and white (n = 1,837; 87%). In terms of educational level, 29% were general dentists, 50% had a certificate degree and 21% had a postgraduate degree (MSc, PhD and/or Postdoc). Almost 70% of the respondents had more than 10 years of clinical experience and did not belong to any COVID-19 risk group. While the vast majority (98,91%) of subjects reported to have knowledge about COVID-19, only 16.8% received any specific training on the use, maintenance, and disposal of personal protective equipment (PPE). The distribution of all sociodemographic, education and work characteristics of the sample is found in Table 1.

Regarding their professional activities, 11.3% of respondents worked in public sector, while 75.2% were in private sector and 13.5% worked in both. Although most practices and workplaces had already adapted to the new national guidelines (n = 1,447; 68.5%), most of them were only partially open or working with restricted access (n = 1,171; 55.5%) (Table 1).

COVID-19 pandemic had a detrimental financial impact on dental practice–only 26.5% (n = 560) respondents reported no income reduction or a reduction lower than 10%, while 37.9% had a reduction between 10–50% and 35.6% had more than 50% of income reduction. Table 2 shows the results of ordinal logistic regression of the financial impact and associated

**Table 1. Sociodemographic, educational, and work characteristics of the sample (n = 2,113).**

| Variable/Category | n | % |
|---|---|---|
| **Sociodemographic** | | |
| **Gender** | | |
| Female | 1,565 | 74.07 |
| Male | 547 | 25.89 |
| Other | 1 | 0.05 |
| **IBGE´s# race classification** | | |
| Indigenous | 0 | 0 |
| Yellow | 126 | 5.96 |
| Brown | 133 | 6.29 |
| Black | 37 | 1.75 |
| White | 1,800 | 85.19 |
| I prefer not to answer | 17 | 0.80 |
| **Age range** | | |
| 20–30 years old | 398 | 18.84 |
| 31–40 years old | 517 | 24.47 |
| 41–50 years old | 684 | 32.37 |
| 51–60 years old | 382 | 18.08 |
| >60 years old | 132 | 6.25 |
| **Leaves in** | | |
| São Paulo | 911 | 43.11 |
| Countryside | 1,202 | 56.89 |
| **COVID-19 Risk group** | | |
| Yes | 521 | 24.66 |
| No | 1,563 | 73.97 |
| I do not know | 29 | 1.37 |
| **Education Characteristics** | | |
| **Higher educational level** | | |
| Dentistry degree | 613 | 29.01 |
| Certificate degree | 1,067 | 50.50 |
| Master's degree | 274 | 12.97 |
| Doctorate degree | 109 | 5.16 |
| Postdoctoral degree | 50 | 2.37 |
| **Time since graduation (clinical experience in years)** | | |
| <4 years | 326 | 15.43 |
| 5–10 years | 297 | 14.06 |
| 11–20 years | 558 | 26.41 |
| >20 years | 915 | 43.30 |
| I do not work in clinical practice | 17 | 0.80 |
| **COVID -19 Education Characteristics** | | |
| **Do you search for information in the Dental Council website?** | | |
| Yes | 1,647 | 77.95 |
| No | 453 | 21.44 |
| Not answered | 13 | 0.62 |
| **Do you acknowledge COVID-19 health safety protocols for clinical practice?** | | |
| Yes | 1,264 | 59.82 |
| Some | 826 | 39.09 |
| None, almost none. | 23 | 1.09 |

*(Continued)*

**Table 1.** (Continued)

| Variable/Category | n | % |
|---|---|---|
| **Have you received any training on the use, maintenance, and disposal of PPE\*?** | | |
| Yes | 355 | 16.80 |
| No | 1,758 | 83.20 |
| **Where?** | | |
| Yes, during undergraduate or postgraduation course | 275 | 13.01 |
| Yes, in my workplace (past or present) | 164 | 7.76 |
| Yes, on my own or internet/ other medias | 861 | 40.75 |
| Yes, more than one of the above | 368 | 17.42 |
| No, I did not take any course | 355 | 16.80 |
| Yes, but did not specify where/how. | 90 | 4.26 |
| **Work characteristics** | | |
| **Where do you work?** | | |
| São Paulo (capital) | 878 | 41.69 |
| Countryside | 1,164 | 55.27 |
| Both | 64 | 3.04 |
| **Public/Private sector** | | |
| Public | 239 | 11.31 |
| Private | 1,589 | 75.20 |
| Both | 285 | 13.49 |
| **Work routine** | | |
| Remains the same/I have adapted to the new reality | 1,447 | 68.51 |
| I am on holidays/not on practice | 221 | 10.46 |
| Working from home | 100 | 4.73 |
| I got fired or resigned | 35 | 1.66 |
| Other | 309 | 14.63 |
| **Workplace operation** | | |
| Open as usual | 606 | 28.69 |
| Partially open/Restricted access | 1,171 | 55.45 |
| Closed | 154 | 7.29 |
| Other | 181 | 8.57 |
| **Income Reduction** | | |
| No income reduction/ less than 10% | 560 | 26.52 |
| Income reduction between 10–50% | 801 | 37.93 |
| More than 50% of income reduction | 751 | 35.56 |
| **Total** | **2,113** | **100** |

\* PPE = Personal Protective Equipment

#IBGE: Brazilian Institute of Geography and Statistics.

variables. Variables such as gender, race, belonging to COVID-19 risk group, educational level and clinical experience did not influence income reduction ($p > 0.05$). However, dentists aged between 31 and 40 years old experienced less financial burden when compared to the younger age group (20–30 years old; OR = 0.77; CI = 0.60–0.98). Also, dentists who worked only in the countryside reported 16% less financial contraction compared to dentists working in the state's capital ($p = 0.041$). Workplaces that were either partially open, had restricted access or were closed showed a negative influence on the reported economic impact when compared to workplaces that were opened as usual ($p < 0.001$). Additionally, dentists from the private sector

**Table 2. Descriptive and ordinal logistic regression analysis of the financial impact (percentage of income reduction) and independent variables.**

| Variable/Category | Financial Impact | | | Ordinal logistic regression analysis | | | |
|---|---|---|---|---|---|---|---|
| | 0–10% | 10–50% | >50% | Unadjusted | | Adjusted | |
| | n (%) | n (%) | n (%) | OR (CI 95%) | p-value | OR (CI 95%) | p-value |
| **Gender** | | | | | | | |
| Female (ref) | 418 (26.73) | 593 (37.92) | 553 (35.36) | 1 | | | |
| Male | 141 (25.78) | 208 (38.03) | 198 (36.20) | 1.04 (0.87–1.24) | 0.648 | - | - |
| **IBGE´s# race classification** | | | | | | | |
| White (ref) | 475 (26.40) | 699 (38.85) | 625 (34.74) | 1 | | | |
| Brown/Black | 42 (24.71) | 66 (38.82) | 62 (36.47) | 1.08 (0.81–1.44) | 0.588 | - | - |
| Yellow | 37 (29.37) | 33 (26.19) | 56 (44.44) | 1.21 (0.85–1.71) | 0.279 | - | - |
| I prefer not to answer | 6 (35.29) | 3 (17.65) | 8 (47.06) | 1.13 (0.43–2.95) | 0.791 | - | - |
| **Age range** | | | | | | | |
| 20–30 years old | 94 (23.62) | 152 (38.19) | 152 (38.19) | 1 | | 1 | |
| 31–40 years old | 148 (28.63) | 210 (40.62) | 159 (30.75) | 0.74 (0.59–0.94) | 0.017* | 0.77 (0.60–0.98) | 0.040* |
| 41–50 years old | 184 (26.94) | 260 (38.07) | 239 (34.99) | 0.85 (0.68–1.07) | 0.190 | 0.89 (0.71–1.12) | 0.405 |
| >50 years old | 134 (26.07) | 179 (34.82) | 201 (39.11) | 0.97 (0.76–1.23) | 0.819 | 1.00 (0.78–1.49) | 0.758 |
| **COVID-19 Risk group** | | | | | | | |
| Yes | 140 (26.87) | 185 (35.51) | 196 (37.62) | 1 | | | |
| No | 414 (26.50) | 603 (38.60) | 545 (34.89) | 0.94 (0.78–1.13) | 0.517 | - | - |
| I do not know | 6 (20.69) | 13 (44.83) | 10 (34.48) | 1.05 (0.53–2.06) | 0.881 | - | - |
| **Education Characteristics** | | | | | | | |
| **Higher educational level** | | | | | | | |
| Dentistry degree | 158 (25.77) | 220 (35.89) | 235 (38.34) | 1 | | | |
| Certificate degree | 280 (26.24) | 407 (38.14) | 380 (35.61) | 0.92 (0.76–1.11) | 0.397 | - | - |
| MSc/PhD/Postdoc | 122 (28.24) | 174 (40.28) | 136 (31.48) | 0.79 (0.63–1.00) | 0.051 | - | - |
| **Time since graduation (clinical experience in years)** | | | | | | | |
| <4 years/not working in clinical practice | 84 (24.49) | 120 (34.99) | 139 (40.52) | 1 | | | |
| 5–10 years | 71 (23.91) | 130 (43.77) | 96 (32.32) | 0.82 (0.61–1.09) | 0.180 | - | - |
| 11–20 years | 155 (27.83) | 219 (39.32) | 183 (32.85) | 0.76 (0.59–0.98) | 0.036* | - | - |
| >20 years | 250 (27.32) | 332 (36.28) | 333 (36.39) | 0.84 (0.67–1.06) | 0.153 | - | - |
| **Work characteristics** | | | | | | | |
| **Where do you work?** | | | | | | | |
| São Paulo (capital) | 218 (24.83) | 322 (36.67) | 338 (38.50) | 1 | | 1 | |
| Countryside | 322 (27.69) | 454 (39.04) | 387 (33.28) | 0.82 (0.70–0.96) | 0.019* | 0.84 (0.72–0.99) | 0.041* |
| Both | 18 (28.13) | 23 (35.94) | 23 (35.94) | 0.87 (0.54–1.39) | 0.567 | 0.92 (0.56–1.49) | 0.743 |
| **Public/Private sector** | | | | | | | |
| Public | 108 (45.19) | 72 (30.13) | 59 (24.69) | 1 | | 1 | |
| Private | 359 (22.61) | 619 (38.98) | 610 (38.41) | 2.43 (1.87–3.15) | <0.001* | 2.41 (1.84–3.14) | <0.001* |
| Both | 93 (32.63) | 110 (38.60) | 82 (28.77) | 1.53 (1.10–2.11) | 0.010* | 1.51 (1.09–2.11) | 0.013* |
| **Workplace operation** | | | | | | | |
| Open as usual | 226 (37.29) | 233 (38.45) | 147 (24.26) | 1 | | 1 | |
| Partially open/ Restricted access | 259 (22.12) | 483 (41.25) | 429 (36.64) | 1.92 (1.59–2.29) | <0.001* | 1.90 (1.58–2.28) | <0.001* |
| Closed | 45 (29.22) | 41 (26.62) | 68 (44.16) | 2.01 (1.43–2.83) | <0.001* | 1.92 (1.35–2.71) | <0.001* |
| Other | 30 (16.57) | 44 (24.31) | 107 (59.12) | 4.18 (3.00–5.82) | <0.001* | 4.15 (2.98–5.79) | <0.001* |
| **Total** | 560 (26.52) | 801 (37.93) | 751 (35.56) | | | | |

IBGE: Brazilian Institute of Geography and Statistics; MSc: Master of Sciences; PhD: Doctor of philosophy; OR = Odds Ratio; CI 95% = Confidence Interval 95%; *p<0.05.

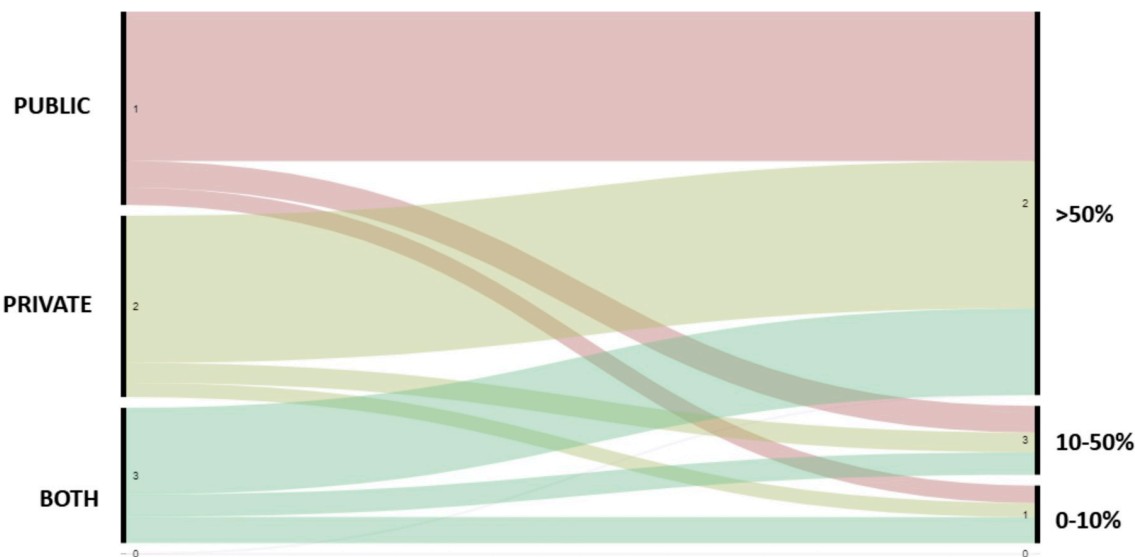

**Fig 1. Alluvial diagram of workplace (public, private or both) and financial impact (income reduction 0–10%; 10–50% and >50%).**

reported suffering 2.41 times greater financially than dentists from the public sector (CI = 1.84–3.14; p<0.001). Likewise, dentists who work in both public and private sectors had higher detrimental impact than those who worked only in the public sector (OR = 1.51; p = 0.013). Fig 1 shows the alluvial diagram showing the distribution of the financial impact among public and private sectors.

Regarding the use of PPE for infection control, the majority of dentists reported using N95 masks, face shields, disposable gowns and shoe covers (Fig 2).

## 4. Discussion

The present study investigated the impact of COVID-19 pandemic on dentists practicing in the state of São Paulo, Brazil, which has more than 639.000 confirmed cases with more than 25.000 deaths [15]. We reported a detrimental financial impact of COVID-19 pandemic on dentists, and provided data on the sociodemographic, educational profile, dental practice characteristics and personal protective equipment (PPE) use by dentists during this time.

The majority (73.50%) of dentists reported a reduction of up to half the income (37.90%) or more than half of the income (35.60%), with this being more prevalent among dentists working in the private sector. This result is in accordance with other studies [16–23], that although with a smaller sample size, have also detected a negative financial impact of COVID-19 pandemic in dentistry worldwide. Dentists aged between 31 and 40 years old experienced less financial burden when compared to the younger age group (20–30 years old; OR = 0.77; CI = 0.60–0.98), which could be due to higher financial estability of older dentists (more years practing dentistry) and/or the fact that the younger dentists may also have other expenditures, such as postgraduate studies costs.

Although the present study focused on dentists, it is important to acknowledge the possible impact of the pandemic on all other professionals working in the dental office, such as dental hygienists, dental assistants. A recent study reported that only 27% of american dental surgeons were able to offer full payment to their employees [22].

Dentists are among the top at-risk occupations for contracting COVID-19 [6]. Concerning the use of PPE for infection control, our data showed that the majority of the respondents

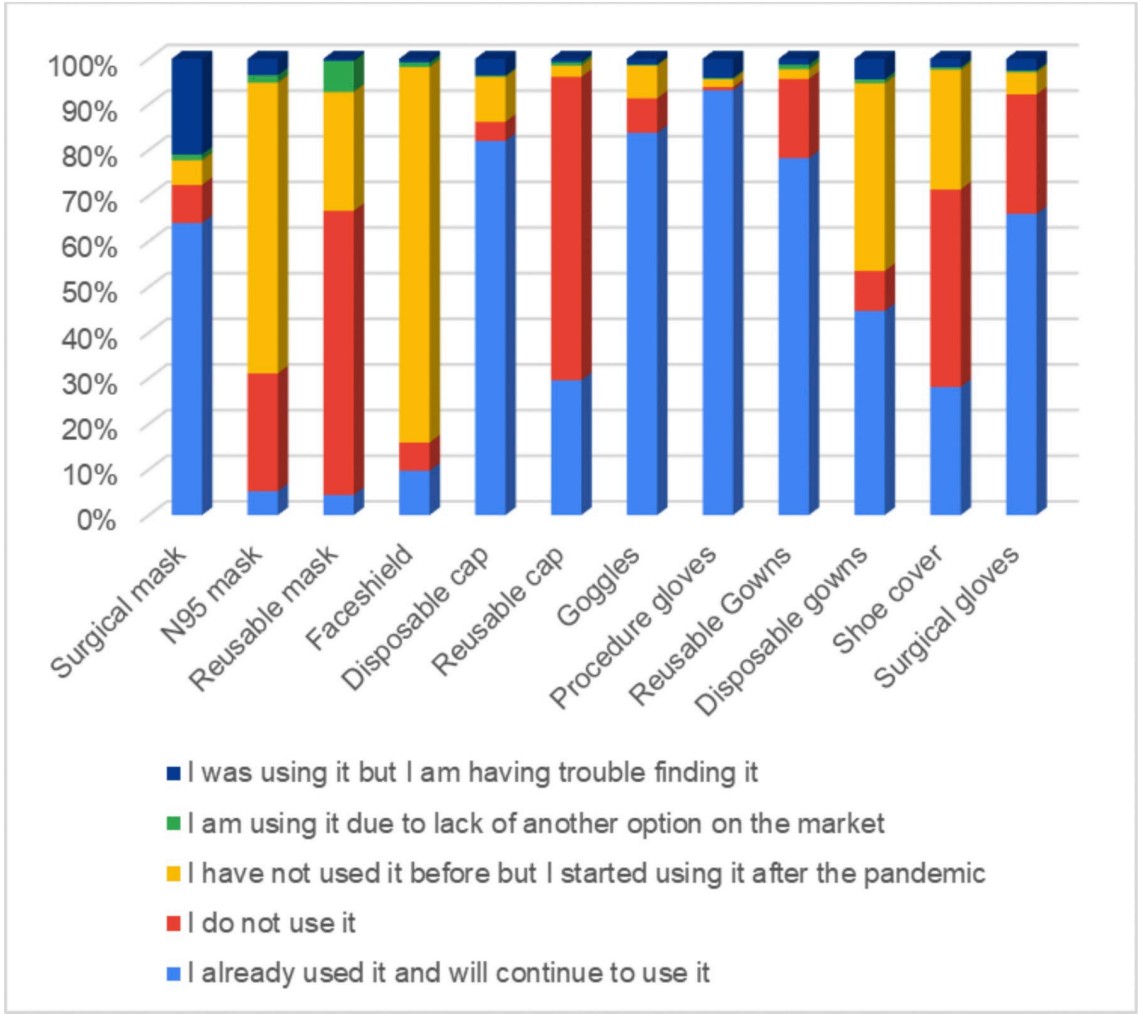

**Fig 2. Distribution of the Personal Protective Equipment (PPE) for infection control used.**

adopt it. It is possible that these add additional costs to the dental practices, justifying a higher financial burden on the private sector when the dentists spend their own resources to purchase these items. Furthermore, dentists also reported difficulties finding PPEs and some resorted to using reusable masks, in agreement with known bottlenecks for accessibility to PPE, that includes supply chain shortages [24, 25].

Unfortunately, 83.2% of the participants in this study reported not having received any specific training to control COVID-19 transmission in the healthcare environment. Although there are plenty of free courses and several ways of continuing education which were widely publicized, such as courses from the Open University in the Brazilian Unified Health System [26] and other free access resources in philanthropic institutions [27], most respondents did not access this information. On the other hand, among dentists that reported having received training on the use, maintenance, and disposal of PPE, 40.75% did so through the internet or other social media. The rate of dissemination of traditional peer reviewed publications, static websites and emails are known to be slower and generally in English language [28].

Dentists of the private network have their remuneration determined by the attendance and procedures performed, either directly, through private offices and clinics, or indirectly,

through the provision of services for agreements and cooperatives. On the other hand, dental professionals from the brazilian public network generally present labour contracts or statutory regime, with fixed remuneration and benefits. Consequently, even with the activities partially paralyzed and the attendance focused only on dental emergency cases, the public sector reorganized its work processes to face the pandemic, expanding the field of action of dental professionals directing them to different functions such as: telemonitoring of confirmed cases of COVID, administrative work, epidemiological inquiries, research, among others [29]. All these factors can explain why the private sector dentists included in the present investigation suffered twice the financial impact when compared to public sector dentists.

The internal validity of this present study needs to be interpreted with caution. The results could have been influenced by the period in which the survey was taken (July 2020), when only 30% of the dental practices were open as usual. Also, dentists who were working full time at the time of the online survey could not had participated and influenced the final results. In relation to the external validity of the present data, especially when extrapolating to other regions in Brazil or other countries in a similar situation, some factors as social and political circumstances, COVID-19 incidence at the time, and local regulations could play an important role in the financial impact.

As the largest country in South America and the fifth largest nation in the world, the COVID-19 pandemic reached different regions in Brazil at different times and proportions [11]. At the time this questionnaire was sent, there were fewer cases of COVID-19 in the countryside, which can explain why dentists from the countryside experienced less impact when compared to dentists from the capital. Populations with a greater number of deaths and COVID-19 confirmed cases will feel the greatest impact. Therefore, the present results could be extrapolated to populations with COVID-19 infection rates similar to those of the state of São Paulo. Furthermore, policy responses to the coronavirus pandemic should also be considered.

It is important to recognize some limitations of the present study, such as the convenience sample, which included dentists who had access to their emails and social media in the period that the questionnaire was released and the short period of data collection. Additionally, the high number of online surveys that are being developed after the start of the COVID-19 pandemic may also have suppressed our ability to reach a better response rate and could have affected the sample size. Further larger population studies are needed in different countries to understand the COVID-19 impact on dentistry.

## 5. Conclusions

COVID-19 pandemic caused major negative impacts on the routine of dentists in the state of São Paulo, Brazil. These professionals reported reduced income by more than 50% and the private dental sector was more affected than the public sector. In addition, about 83% reported not having received any specific training to control the transmission of coronavirus in the health area. This data added to the difficulty in finding PPE is extremely worrying since the dentist is among the occupations at greatest risk for the contamination of COVID-19. We hope that this study will help dentists and other health professionals to better understand the consequences of the disease and thus search more information's about care against COVID-19.

## Supporting information

**S1 Table. STROBE statement—checklist of items that should be included in reports of cross-sectional studies.**
(DOCX)

**S2 Table. Questionnaire.** (original language: Brazilian Portuguese).
(DOC)

**S3 Table. Translated questionnaire.** (English).
(DOC)

**S4 Table. Data set.**
(XLSX)

## Acknowledgments

We would like to thank the members of the center for study, research and practice in PHC networks at the Hospital Israelita Albert Einstein for their helpful suggestions.

## Author Contributions

**Conceptualization:** Tatiane Fernandes Novaes, Maisa Camillo Jordão, Danielle da Costa Palacio, Debora Heller.

**Data curation:** Maisa Camillo Jordão, Danielle da Costa Palacio, Debora Heller.

**Formal analysis:** Tatiane Fernandes Novaes, Maisa Camillo Jordão, Isabel Cristina Olegário, Daniele Boina de Oliveira, Veranika Ushakova, Alexander Birbrair, Danielle da Costa Palacio, Debora Heller.

**Funding acquisition:** Alexander Birbrair.

**Investigation:** Carlos Felipe Bonacina, André Oswaldo Veronezi, Danielle da Costa Palacio, Debora Heller.

**Methodology:** Tatiane Fernandes Novaes, Maisa Camillo Jordão, Carlos Felipe Bonacina, André Oswaldo Veronezi, Danielle da Costa Palacio, Debora Heller.

**Supervision:** Danielle da Costa Palacio, Debora Heller.

**Validation:** Tatiane Fernandes Novaes, Maisa Camillo Jordão, Danielle da Costa Palacio, Debora Heller.

**Visualization:** Tatiane Fernandes Novaes, Maisa Camillo Jordão, Danielle da Costa Palacio, Debora Heller.

**Writing – original draft:** Tatiane Fernandes Novaes, Maisa Camillo Jordão, Carlos Felipe Bonacina, André Oswaldo Veronezi, Isabel Cristina Olegário, Daniele Boina de Oliveira, Danielle da Costa Palacio.

**Writing – review & editing:** Carlos Ariel Rodrigues de Araujo, Veranika Ushakova, Alexander Birbrair, Debora Heller.

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
