## [Decision Letter · Decision Letter 0]

16 Jun 2021

PONE-D-20-40253

COVID-19 pandemic impact on dentists in Latin America’s epicenter: São-Paulo, Brazil

PLOS ONE

Dear Dr. Heller,

Thank you for submitting your manuscript to PLOS ONE. After careful consideration, we feel that it has merit but does not fully meet PLOS ONE’s publication criteria as it currently stands. Therefore, we invite you to submit a revised version of the manuscript that addresses the points raised during the review process.

We look forward to receiving your revised manuscript.

Kind regards,

Endi Lanza Galvão

Academic Editor

PLOS ONE

Journal Requirements:

Reviewers' comments:

Reviewer's Responses to Questions

**Comments to the Author**

1. Is the manuscript technically sound, and do the data support the conclusions?

Reviewer #1: Yes

Reviewer #2: Yes

Reviewer #3: Partly

Reviewer #4: Yes

2. Has the statistical analysis been performed appropriately and rigorously? 

Reviewer #1: Yes

Reviewer #2: Yes

Reviewer #3: Yes

Reviewer #4: Yes

3. Have the authors made all data underlying the findings in their manuscript fully available?

Reviewer #1: Yes

Reviewer #2: Yes

Reviewer #3: Yes

Reviewer #4: Yes

4. Is the manuscript presented in an intelligible fashion and written in standard English?

Reviewer #1: Yes

Reviewer #2: Yes

Reviewer #3: Yes

Reviewer #4: Yes

5. Review Comments to the Author

Reviewer #1: This is an interesting manuscript that provides relevant information about the economic impact of the COVD epidemic for dentists in a large Brazilian city. The relevance of this manuscript highlights the need for more information about individual protection methods for the performance of dentists.

Reviewer #2: This cross sectional study was done in Sao Paulo Brazil to evaluate the impact of COVID 19 among dentists. A good study to evaluate an important dimension. I have some observation mentioned below

Abstract

Line 37 : How many dentists were approached?

Result

Line 136 : Pl recheck n 2348-2113 = 235

Line 140/141 : please use uniform nomenclature in table and text

Conclusion :

Please try to have a reflection of objectives

Reviewer #3: I declare that I have no conflict of interest with the authors and the institution by which the work was developed. The work is in ethical compliance.

The article brings an interesting theme in view of the various impacts that the pandemic has brought to the population. Health professionals are part of the front line and special attention must be paid to all of them. However, there are methodological limitations and some unclear points.

The following are some doubts and questions about the clarity of the data:

Summary:

1- As the study recruited only dentists from the state of São Paulo, and only 2.26% of them participated in the survey, it is wrong to conclude “This study provides evidence of the negative impact of the COVID-19 pandemic on dentistry in Brazil.”

Introduction:

2- The authors claim to have followed the standards proposed by the Strengthening the Reporting of Observational Studies in Epidemiology (STROBE), however, they do not present the study hypothesis as guided by the guide.

Methodology:

3- Was it considered only active registration or were dentists who deactivated their registration also considered?

4- In which month / period of time did the study take place? According to STROBE, this information must come in the methodology section.

5- How long was the data collected for? This should also come in the methodology and not in the results section.

6- In relation to dentists who were not working in dentistry, but registered in CROSP (active or inactive), how was this controlled?

7- The authors state in the supplementary document with the completed STROBE that the item “variables” in the methodology section does not apply to the study. Because? Since the authors work with categorical variables. Clearly define exposures and potential confounders.

8- How much does the recruited sample represent the population of dentists in the state of SP?

Results:

9- Why did the data collection only last 8 days?

10- Why were the variables from the “COVID -19 Education Characteristics” domain of the questionnaire not used in the regression? ”

11- Could the sample size be a limitation?

12- Talk about the internal and external validity of the data.

Discussion

13- Why did older dentists suffer less from the financial impacts compared to younger ones?

Conclusions

14- The authors conclude for the Brazilian population only 2.26% of dentists in the state of São Paulo.

The conclusions do not answer the objective objectively.

Reviewer #4: This study aimed to assess the impact of the COVID-19 pandemic on dentists in the State of São Paulo. Through a questionnaire, via Google form, the authors sought information about changes in dentists' routine during the pandemic. Important evidence was generated confirming the negative impact of the pandemic on the income of these professionals. In addition, interesting results about the difficulty in purchase PPE and the lack of specific training to control coronavirus transmission were found. I believe this study brings clarity about the need to improve the dissemination of information about care against coronavirus. I congratulate the authors for this work and suggest minor changes before publication.

1. Introduction section - Page 3 - line 55: The infection is caused by SARS-CoV 2 which causes COVID-19 disease. Correct the sentence: “COVID-19 infected cases have since grown exponentially…”

2. Methods section: I suggest informing that research was carried out with professionals from the State of São Paulo, at first seemed to be only with professionals from the capital.

3. I consider the results on training to control coronavirus transmission and disposal of PPE of the most important in the study. Therefore, I suggest including in the summary and conclusion as one of the main results found.

4. Conclusion section: The authors' conclusion was: “We presented evidence of the negative impact of COVID-19 pandemic on dentists in the State of São Paulo, Brazil. Brazilian dentists reported an income reduced by more than 50% and the dental private sector shouldering twice the burden of those in the public sector. While these are unprecedented challenging times, we hope this study will help dental and other health care professionals to better understand the consequences of disease in dental settings and strengthen preparedness throughout the dental health care system.”

I suggest making the conclusion more clearer and objective.

Suggestion: COVID-19 pandemic caused major negative impacts on the routine of dentists in the state of São Paulo, Brazil. These professionals reported reduced income by more than 50% and the private dental sector was more affected than the public sector. In addition, about 83% reported not having received any specific training to control the transmission of coronavirus in the health area. This data added to the difficulty in finding PPE is extremely worrying since the dentist is among the occupations at greatest risk for the contamination of COVID-19. We hope that this study will help dentists and other health professionals to better understand the consequences of the disease and thus search more information’s about care against COVID-19.

6. PLOS authors have the option to publish the peer review history of their article (what does this mean?). If published, this will include your full peer review and any attached files.

Reviewer #1: No

Reviewer #2: No

Reviewer #3: No

Reviewer #4: **Yes: **Tamires Szeremeske Miranda

---

## [Author Response · Author response to Decision Letter 0]

8 Jul 2021

São Paulo, 7/08/2021

Dear Dr. Endi Lanza Galvão,

The authors of the manuscript entitled “COVID-19 pandemic impact on dentists in Latin America’s epicenter: São-Paulo, Brazil” have carefully evaluated the reviewers’ comments and express their thanks for their constructive critique. All suggestions and questions have been addressed in the revised manuscript. The changes suggested by the reviewers have been incorporated in the manuscript and are listed below. We would like to point out that we made additional modifications and corrections in order to improve the clarity of the manuscript. 

Again, we would like to thank the reviewers for theirs constructive suggestions. We hope that the changes meet with your approval and make this manuscript acceptable for publication. 

Sincerely,

Dr. Debora Heller

July 8, 2021

We appreciate the expert feedback provided by the reviewers. It has enabled us to increase the clarity and quality of the manuscript. We incorporated the requested major changes. Below we present a point-by-point response to clearly identify the changes made to the text (changes also marked in the manuscript). 

Journal Requirements:

Response: We checked all style requirements and modified the manuscript accordantly.

Response: We included captions for all Supporting Information files at the end of the manuscript, and updated all in-text citations to match accordingly.

Response to the reviewers’ comments

Reviewer #1

This is an interesting manuscript that provides relevant information about the economic impact of the COVD epidemic for dentists in a large Brazilian city. The relevance of this manuscript highlights the need for more information about individual protection methods for the performance of dentists.

Response: We thank the reviewer for the positive feedback on our manuscript and are glad to learn that our message was clear to him/her. The feedback was valuable to us and also taken into consideration while preparing the revisions.

Reviewer #2

This cross sectional study was done in Sao Paulo Brazil to evaluate the impact of COVID 19 among dentists. A good study to evaluate an important dimension. I have some observation mentioned below

Response: Thank you for the positive feedback and critique. 

Abstract

Line 37: How many dentists were approached?

Response: A semi-structured questionnaire was sent via e-mail to 93.280 dentists registered in the Dental Council of São Paulo (CROSP). This information has been added to the abstract (Line 35)

Result

Line 136: Pl recheck n 2348-2113 = 235

Response: Thank you for your correction. This information was corrected in the text. In fact, 101 participants refused to participate after accessing the informed consent; the others refused before that, totalizing 235.

Line 140/141: please use uniform nomenclature in table and text

Response: Uniform nomenclatures have been added to the table and text as suggested.

Conclusion:

Please try to have a reflection of objectives

Response: This critique is in agreement with the other reviewers and was addressed by making the conclusion more clear and objective (Lines 257-267).

Reviewer #3

I declare that I have no conflict of interest with the authors and the institution by which the work was developed. The work is in ethical compliance.

The article brings an interesting theme in view of the various impacts that the pandemic has brought to the population. Health professionals are part of the front line and special attention must be paid to all of them. However, there are methodological limitations and some unclear points.

Response: Thank you for your feedback. All critiques and suggestions have been addressed in the revised manuscript. 

The following are some doubts and questions about the clarity of the data:

Summary:

1- As the study recruited only dentists from the state of São Paulo, and only 2.26% of them participated in the survey, it is wrong to conclude “This study provides evidence of the negative impact of the COVID-19 pandemic on dentistry in Brazil.”

Response: To address this reviewer’s critiques we have replaced this statement as follows: “This study provides evidence of the negative impact of the COVID-19 pandemic on the routine of dentists in the state of São Paulo, Brazil.”

Introduction:

2- The authors claim to have followed the standards proposed by the Strengthening the Reporting of Observational Studies in Epidemiology (STROBE), however, they do not present the study hypothesis as guided by the guide.

Response: The null hypothesis of the study is that the COVID-19 pandemic would not impact the routine of dentists in the state of São Paulo. The study's alternative hypothesis is that the COVID-19 pandemic would impact the routine of dentists in the state of São Paulo. This information has been added to the text.

Methodology:

3- Was it considered only active registration or were dentists who deactivated their registration also considered?

Response: Only active registrations were considered. This information has been added to the text.

4- In which month / period of time did the study take place? According to STROBE, this information must come in the methodology section.

Response: The present study took place from June 25, 2020 until July 02, 2020. This information has been added to the methodology section (Lines 111-112).

5- How long was the data collected for? This should also come in the methodology and not in the results section.

Response: The data was collected for 8 days. This information has been added to the methodology section as suggested (Line 111).

6- In relation to dentists who were not working in dentistry, but registered in CROSP (active or inactive), how was this controlled?

Response: Only dentists with active registrations were contacted by email and included in this study. This information has been added to the text. 

7- The authors state in the supplementary document with the completed STROBE that the item “variables” in the methodology section does not apply to the study. Because? Since the authors work with categorical variables. Clearly define exposures and potential confounders.

Response: All independent variables were considered in the regression model. So, the adjusted model was presented, describing the Confidence Interval and Odds Ratio values. This information was added in the Statistical analyses section. The STROBE checklist was also corrected.

8- How much does the recruited sample represent the population of dentists in the state of SP?

Response: From total of 93.280 dentists with active registration in the Regional Dental Council of Sao Paulo, 2.113 participated in this study. This represents 2.3% of the population of dentists in the state of SP. This information has been added to the text and addressed accordantly. 

Results:

9- Why did the data collection only last 8 days?

Response: This study took place in June 2020, when São Paulo was the epicenter of the COVID-19 Pandemic in Latin America. In order to gather reliable information and help the dental community, we decided to limit the questionnaire for 8 days. The response rate was higher in the first three days (80% of total responses), decreasing to 7, 6, 4, 2 and 1% on each of the following day. We believe that even a longer period of data collection would not impact in higher participants number in the present survey, since not many responses were obtained after 7-8 days. 

10- Why were the variables from the “COVID -19 Education Characteristics” domain of the questionnaire not used in the regression? ”

Response: The primary outcome of the present study was the COVID-19 financial impact. As the presence of training/education on COVID aspects does not have a plausible association with financial impact, the authors decided not to include this independent variable in the regression analysis. However, the descriptive report of this variable has been added into this paper to describe to the reader that many dentists at the time of the present study did not have any previous educational support in dealing with COVID pandemic.

11- Could the sample size be a limitation?

Response: From total of 93.280 dentists with active registration in the Regional Dental Council of Sao Paulo, 2.113 participated in this study. The responses could be influenced by the fact that the participants were not randomly selected (convenience sample) and the results are only representative of those who answered the online survey. We agree that this could be a study limitation and added to the discussion section. 

12- Talk about the internal and external validity of the data.

Response: We have added a brief discussion in relation to the internal and external validity of the data to our discussion section. 

The internal validity of this present study needs to be interpreted with caution. The results could have been influenced by the period in which the survey was taken (July 2020), when only 30% of the dental practices were open as usual. Also, dentists who were working full time at the time of the online survey could not had participated and influenced the final results. In relation to the external validity of the present data, especially when extrapolating to other regions in Brazil or other countries in a similar situation, some factors as social and political circumstances, COVID-19 incidence at the time, and local regulations could play an important role in the financial impact. 

Discussion

13- Why did older dentists suffer less from the financial impacts compared to younger ones?

Response: There could be many reasons for this, such as more stability of older dentist (more years practicing dentistry) and or the fact that the younger dentists may also have other expenditures, such as postgraduate studies costs. This has now been addressed in the discussion. 

Conclusions

14- The authors conclude for the Brazilian population only 2.26% of dentists in the state of São Paulo.

The conclusions do not answer the objective objectively.

Response: We believe that the conclusions of the paper were not appropriately presented. We took this critique into strong consideration and modified conclusions as suggested making it clearer and objective.

Reviewer #4

This study aimed to assess the impact of the COVID-19 pandemic on dentists in the State of São Paulo. Through a questionnaire, via Google form, the authors sought information about changes in dentists' routine during the pandemic. Important evidence was generated confirming the negative impact of the pandemic on the income of these professionals. In addition, interesting results about the difficulty in purchase PPE and the lack of specific training to control coronavirus transmission were found. I believe this study brings clarity about the need to improve the dissemination of information about care against coronavirus. I congratulate the authors for this work and suggest minor changes before publication.

Response: Thank you for the positive feedback.

1. Introduction section - Page 3 - line 55: The infection is caused by SARS-CoV 2 which causes COVID-19 disease. Correct the sentence: “COVID-19 infected cases have since grown exponentially…”

Response: Corrected as suggested.

2. Methods section: I suggest informing that research was carried out with professionals from the State of São Paulo, at first seemed to be only with professionals from the capital.

Response: Replaced as suggested.

3. I consider the results on training to control coronavirus transmission and disposal of PPE of the most important in the study. Therefore, I suggest including in the summary and conclusion as one of the main results found.

Response: Thank you for the suggestion. We have now added this to the summary and conclusions.

4. Conclusion section: The authors' conclusion was: “We presented evidence of the negative impact of COVID-19 pandemic on dentists in the State of São Paulo, Brazil. Brazilian dentists reported an income reduced by more than 50% and the dental private sector shouldering twice the burden of those in the public sector. While these are unprecedented challenging times, we hope this study will help dental and other health care professionals to better understand the consequences of disease in dental settings and strengthen preparedness throughout the dental health care system.”

I suggest making the conclusion more clearer and objective.

Suggestion: COVID-19 pandemic caused major negative impacts on the routine of dentists in the state of São Paulo, Brazil. These professionals reported reduced income by more than 50% and the private dental sector was more affected than the public sector. In addition, about 83% reported not having received any specific training to control the transmission of coronavirus in the health area. This data added to the difficulty in finding PPE is extremely worrying since the dentist is among the occupations at greatest risk for the contamination of COVID-19. We hope that this study will help dentists and other health professionals to better understand the consequences of the disease and thus search more information’s about care against COVID-19.

Response: We agree that the conclusions of the paper were not appropriately presented. We took this critique into strong consideration and modified conclusions as suggested making it clearer and objective.

---

## [Editor Report · Decision Letter 1]

30 Jul 2021

COVID-19 pandemic impact on dentists in Latin America’s epicenter: São-Paulo, Brazil

PONE-D-20-40253R1

Dear Dr. Heller,

We’re pleased to inform you that your manuscript has been judged scientifically suitable for publication and will be formally accepted for publication once it meets all outstanding technical requirements.

Kind regards,

Endi Lanza Galvão

Academic Editor

PLOS ONE
---

## [Editor Report · Acceptance letter]

18 Aug 2021

PONE-D-20-40253R1 

COVID-19 pandemic impact on dentists in Latin America’s epicenter: São-Paulo, Brazil 

Dear Dr. Heller:

I'm pleased to inform you that your manuscript has been deemed suitable for publication in PLOS ONE. Congratulations! Your manuscript is now with our production department. 

Kind regards, 

on behalf of

Dr. Endi Lanza Galvão 

Academic Editor

PLOS ONE